# Justifying Euthanasia: A Qualitative Study of Veterinarians’ Ethical Boundary Work of “Good” Killing

**DOI:** 10.3390/ani13152515

**Published:** 2023-08-04

**Authors:** Marc J. Bubeck

**Affiliations:** Faculty of Health Sciences Brandenburg, Universität Potsdam, 14469 Potsdam, Germany; marc.bubeck@uni-potsdam.de

**Keywords:** veterinary ethics, qualitative research, ethical boundary work, human–animal relationships, euthanasia, companion animals, farm animals, veterinary humanities, sociology

## Abstract

**Simple Summary:**

(1) This study focuses on the ethical challenges veterinarians face when euthanizing animals, the act of ending an animal’s life to relieve its suffering. Unlike other healthcare professionals, veterinarians are often required to perform euthanasia as part of their work. How veterinarians determine what constitutes a “good” killing, in a normative sense, needs to be explored. (2) 17 interviews with veterinarians were conducted and analyzed in detail. (3) The study found that veterinarians have different perspectives on what they consider ethically acceptable regarding euthanasia. They distinguish between farm animals and companion animals. Economic and emotional factors also influence their decisions. Ethical boundary work describes how veterinarians define what they consider normatively legitimate in these areas of veterinary medicine. (4) In conclusion, this study shows that veterinarians face difficult decisions and use ethical boundary work to meet these challenges. They must balance sometimes conflicting interests and adapt to multiple situations. By understanding the complexity of ethical boundary work, we can better understand the moral aspects of veterinary practice. This knowledge can improve veterinary care and help veterinarians make ethical decisions that benefit both animals and society.

**Abstract:**

(1) Veterinarians are regularly required to euthanize their “objects of care” as part of their work, which distinguishes them from other healthcare professionals. This paper examines how veterinarians navigate the ethical tensions inherent in euthanasia, particularly the collision between the routine practice of killing animals within their profession and the broader social and moral implications. (2) Using the sociological concept of ethical boundary work as a theoretical framework, this research observes how veterinarians draw boundaries by positioning their euthanasia practices on the ethical “good” spectrum. A grounded theory study of 17 qualitative interviews with veterinarians was conducted. (3) The findings highlight differences in ethical boundary work within veterinary medicine, particularly in the distinction between farm animals and companion animals. Economic and emotional reasoning play differing roles in explanation and justification. Ethical boundary work is a tool for distinguishing normative frameworks in different areas of veterinary medicine. (4) In conclusion, veterinarians grapple with the realities of an imperfect world and often rely on boundary work to assert diverse interests and navigate multiple contexts. By exploring the complexities of ethical boundary work, this study contributes to a more comprehensive understanding of the moral landscape within veterinary practice.

## 1. Introduction


*“If you say you’re a vet, then very quickly you get, ‘Yes, but then you have to euthanise animals; I couldn’t do that.’ I think, ‘Yes, you can!’ as I said, with a reasonably justified decision.”*
[Dr. Fischer] (The names mentioned in the text have been pseudonymized to protect the interviewees’ privacy).

In modern society, veterinary medicine is crucial as a healthcare profession [1,2]. Trusted by the public as the primary medical caregivers for animals, veterinarians carry a significant ethical burden in their practice. However, despite society’s expectations of healthcare professions, veterinarians also kill the animals they care for. The killing of animals takes many forms and presents complex ethical challenges in veterinary medicine because of the various uses of animals, the involvement of multiple stakeholders, and the broad range of responsibilities of the profession. Euthanasia is a controversial issue involving a “*reasonably justified decision*” (Dr. Fischer) by veterinarians. The evolving normative self-image of veterinarians has increased awareness of ethical considerations in their professional roles, resulting in the emergence of veterinary ethics as a distinct discipline [3,4].

As a problem-oriented discipline, veterinary ethics plays a critical role in developing explanations to evolving ethical issues and reflecting on the nature of ethical problems in veterinary medicine [3]. However, it is essential to note that “*veterinarians’ real-life problems can only be addressed and solved by considering the contextual complexity and requirements of the veterinary profession that result from the relationship between the animal, the client, and the veterinarian*” [3] (p. 481). For example, the animal’s best interest is often seen as the profession’s main principle. Still, veterinarians may have conflicting duties to other stakeholders that go beyond the animal’s best interest [3,5,6,7,8,9]. For example, in animal agriculture and food production, veterinarians have multiple responsibilities, such as animal welfare, food safety, and public health, and must navigate the competing demands of various stakeholders. And that is just in farm animal medicine. Because of the many ways animals are used in society, the moral infrastructure of veterinary medicine is complex. Veterinarians must manage these manifold relationships between animals, clients, and society. Veterinary ethics requires a thorough understanding of context-specific responsibilities in these different work areas and the ability to navigate complex ethical dilemmas in these contexts [2,3].

The literature review on euthanasia in veterinary medicine focuses on the ethical challenges veterinarians face [3,9,10,11,12,13]. In their training, prospective veterinarians learn some techniques and justifications for euthanasia [14,15] to incorporate this aspect of their work into their professional identity [16,17,18]. Different techniques and methods are used to kill an animal, depending on the context and stakeholders involved. This has led to an understanding of veterinary practice as a triad (veterinarian, client, and animal) [7,8,12,19]. Accordingly, veterinarians have different “*objects of care*” [20] (p. 60). The meaning and relevance of these responsibilities may vary between fields (e.g., companion [19,21], farm [20,22,23], and laboratory animals [2]). In addition, the shift to companion animal care has changed the character of the veterinary profession [24,25] and highlights the complex and changing nature of human-animal relationships [26,27]. Meaning and practice depend on specific contexts and are ambivalently debated within the profession and society [28,29]. For example, euthanasia can be seen as a great privilege of the profession [30,31]. This is in contrast to human medicine, where euthanasia has varying degrees of legality and is highly controversial. Or it can be seen as a significant ethical and emotional burden [11,32,33]. In some cases, veterinarians may even refuse euthanasia on moral grounds [34].

Recent research highlights the ethical dimensions and philosophical assumptions underlying euthanasia and underscores the need for greater awareness and understanding of the complexities involved [23,35]. One aspect of this discussion is the difference between a veterinarian’s personal morality on the one hand and legal and organizational requirements or expectations (e.g., guidelines) on the other [35]. Persson et al. argue that a “*lack of awareness of the underlying philosophical assumptions regarding possible understandings of euthanasia*” (p. 1) is a central issue. Therefore, for this study veterinarians from different fields were interviewed to explore this multi-contextuality and moral infrastructure [12,28,36]. It highlights how they address ethical boundaries within the profession.

This article focuses on euthanasia as a prime example of the ethical boundary work veterinarians must navigate in practice. The sociological concept of ethical boundary work [2,37] is used to understand how normative boundaries are made within the profession. This involves negotiating and delineating the ethical boundaries of what constitutes “good” killing for veterinarians. For example, in the case of animal experimentation, veterinarians must balance the ethical considerations of animal welfare with the scientific value of the research [2]. Although often viewed as a humane way to end an animal’s suffering, euthanasia challenges societal expectations and presents ethical dilemmas for veterinarians. The decision to euthanize an animal must be based on sound medical and ethical considerations, considering the animal’s quality of life, the client’s wishes, and the veterinarian’s professional responsibilities. The nature of these requirements may vary depending on the area of veterinary medicine. For example, in the case of companion animals, veterinarians must consider the animal’s quality of life and the client’s emotional attachment. In contrast, in the case of farm animals, veterinarians must balance the animal’s welfare with the farmer’s economic interests.

To further explore the moral infrastructure of the profession, this paper focuses on euthanasia practices in companion and farm animal medicine and the ethical boundary work veterinarians do. By comparing approaches, it is possible to investigate how veterinarians navigate normative expectations and ethical boundaries in and between these two fields. The analysis sheds light on the weighting of medical, economic, and emotional considerations in the legitimacy of “good” killing. It highlights potential conflicts between veterinarians and their clients, as well as among veterinarians.

Working with ethical boundaries is critical for veterinarians to navigate the moral complexities of modern human–animal relationships. Studying how ethical boundary work is practiced in veterinary medicine can provide insight into the challenges and tensions that arise at these boundaries. The findings show that it is worth looking at boundaries because they are spaces where normative negotiations occur. At the same time, because boundaries are not rigid but fluid, much is in motion at the boundaries. This study contributes to a broader professional and societal discourse, facilitates a reassessment of the legitimacy and morality of euthanasia, and draws attention to the sociohistorical construction of animal categories and boundaries within this discussion.

## 2. Theoretical Framework: Ethical Boundary Work

This article uses the sociological concept of ethical boundary work to study veterinarians’ strategies to differentiate and legitimize their killing practices in different contexts. This is a concept developed by Wainwright et al. [37] and used in a number of empirical studies to understand how professionals define and defend their work in areas of ethical controversy. For example, the development of in vitro meat [38] and in vitro fertilization [39,40], and the use of animals in animal research [2,41]. Common to these studies is an orientation toward situated ethics, which refers to how ethical practices are embedded in sociocultural settings [2,42]. For this reason, this concept is helpful for empirically examining the profession’s moral infrastructure.

In an ethnographic study of embryonic stem cell research, Wainwright et al. [37] used 15 interviews with scientists to describe the boundaries of their ethical scientific activities and how they draw them. In the study, they showed how scientists delineated a normatively positive “*ethical space*” (p. 744) for reflecting on and justifying their actions. In doing so, they drew on Gieryn’s [43,44] concept of boundary work, which explores the rhetorical demarcation between science and “*non-science*”. Science, according to Gieryn, is not a static entity but a negotiated practice that distinguishes itself from “*non-science*” by drawing boundaries. This boundary work serves a purpose: “*Boundary-work occurs as people contend for, legitimate, or challenge the cognitive authority of science (…). Pragmatic demarcations of science from non-science are driven by a social interest in claiming, expanding, protecting, monopolizing, usurping, denying, or restricting the cognitive authority of science*” [45] (p. 405). He further describes “*repertoires*” of boundary work. These patterns explain how such work is applied and how each historical episode is related to the construction of boundaries. Moreover, its function is to preserve the social privileges of scientists. Ethics was not considered part of science.

Wainwright et al. [37] found similar but contradictory conclusions, therefore the need for ethical boundary work. They find a process of social demarcation that “*simultaneously serves to define and defend the work of scientists involved in ethically controversial science*” (p. 745). Thus, “*ethics have become another line of demarcation, not so much from ‘non-science’ as from ‘less ethical’ positions*” (p. 745). This allows scientists to present “*themselves as ethical, as well as expert, actors*” [39] (p. 1126). That is why ethics have become part of science. Instead of enhancing and strengthening the prestige and status of science, this form of boundary work reinforces the authority of “*non-science*” (for example, regulatory agencies).

Anderson and Hobson-West [2] build on this and extend ethical boundary work into a spatial realm. They examine how veterinarians draw discursive boundaries between physical spaces, such as the general practice clinic and the animal research laboratory. This spatial differentiation of the multitude of “*objects of care*” [20] highlights the different ways ethical boundary work is enacted. In their study, topography is brought to the fore through three fundamental mechanisms: notions of scale, the scope of authority, and consistency of care. These mechanisms contribute to differentiating the veterinary care required in multiple spaces. Furthermore, this spatial dimension of ethical boundary work plays a crucial role in shaping the moral image of veterinarians’ killing work as a legitimate and integral aspect of their professional practice. Understanding how veterinarians create and maintain a positive ethical perception of their killing work as a site of professional veterinary work is a critical question in this laboratory.

By providing a theoretical lens through which to understand veterinarians’ practices and attitudes in different contexts, the concept of ethical boundary work will guide the empirical analysis.

## 3. Materials and Methods

This study uses a constructivist grounded theory (GT) approach [46] to comprehensively examine the ethical boundary work within the various fields of veterinary medicine. It is an integral part of a broader sociological project that explores the interconnectedness of killing practices and veterinary professionalism. The primary goal of this project is to construct a robust “*grounded theory*” [46] that builds on empirical data and provides a comprehensive understanding of the ethical considerations, practical implications, and emotional complexities associated with the act of killing in the veterinary profession.

To achieve this, 17 qualitative interviews were conducted in an iterative research process with veterinarians working in different areas of the profession. In particular, the analysis of this article focuses on the demarcation between companion and farm animal practices. The methodological approach of this study embodies, on the one hand, an empirical openness characterized by the absence of predetermined boundaries and, on the other hand, an interactive research process. As such, it allows the identification of notable differences in motives, methods, locations, and fields related to killing practices. These findings guided deliberate decisions regarding further recruitment of interviewees, using theoretical sampling methods to ensure a narrower focus within the study [46].

The following sections of this paper follow the COREQ checklist (Consolidated Criteria for Reporting Qualitative Research) [47] to provide a comprehensive account of the research method employed in this study. Intersubjectivity and transparency of the research results should be created through a history of the research process. The following description assumes research as practice, i.e., it is based on a constructivist understanding of knowledge (production).

### 3.1. Developing a Study Design

The study design for this research project was developed using Maxwell’s interactive approach [48], which includes two frameworks: the “top triangle”, consisting of objectives, a conceptual framework, and research questions and the “bottom triangle”, consisting of methods, validity, and research questions. These frameworks were central to developing a research question-oriented design that ensured its appropriateness for the phenomenon under study. In addition, reflexivity was included as a quality criterion for qualitative research [46,49,50].

This exploratory research project addresses the paucity of sociological research in veterinary medicine, particularly with a comparative research design [1,24]. Empirical sociological research on killing in veterinary medicine focuses on one setting or field, such as small animal clinics [17,18,19]. Therefore, GT’s constant comparison [51] and theoretical sampling [46,51,52] contribute to comparative analysis and theory development despite a smaller sample size (compared to quantitative approaches). It is important to note that representativeness is not a quality criterion in GT research [46,50,52]. The design developed allows us to study the boundary work of veterinarians.

Reflexivity played a vital role in the design process for two reasons: First, the researcher’s dual training as a veterinarian and a sociologist provided a unique perspective that allowed a deeper understanding of the subject matter. Second, the researcher’s previous experience with qualitative research, particularly in his master’s thesis in sociology, served as a foundation for developing the current research interest and design. The master’s thesis involved interviewing veterinary students to explore their experiences and perspectives on their studies. Analyzing the integration of killing into their professional identity was a key finding of the study. The results from this previous research and the author’s experiences have informed and influenced the development of the current research project.

### 3.2. Recruitment and Participants

Recruitment played a crucial role in addressing the research question by not pre-determining the boundaries of veterinary medicine. To achieve this, the method of theoretical sampling was used [46]. The recruitment process involved multiple approaches, including snowball sampling, initial contact after web searches of websites and social media, and the use of distribution lists (including three state veterinary associations). It was found that direct contact yielded better results than using email distribution lists (which resulted in a near-zero response rate). As a result, the primary focus was on direct contact initiated by web searches, especially on professional websites, social media, and snowballing.

In GT, data collection, analysis, and theory building form an iterative, intertwined process rather than one single step. This iterative approach allows us to explore contrasting cases through theoretical sampling, such as examining the ethical evaluation of killing by contrasting individual animal treatment with collective herd care. The sampling process also considered variations in “*methods*” (chemical and mechanical) and “*motives*” (suffering avoidance, nutrition, and knowledge production) [10,28].

Seventeen interviews were conducted with veterinarians in Germany from various backgrounds, carefully selected to ensure a comprehensive exploration of the research topic. Differentiation was achieved by considering factors such as field of work (8 small animals, five equines, six farm animals, three laboratory animals, two veterinary offices, one exotic animal), years of experience (four junior (<5 years), eight intermediate (between >5 and <10 years), and four senior (>10 years)), and gender (fourteen females, three males). Almost all individuals had a “Dr. med. vet.” (PhD) except for three (one graduate, two postgraduate), and four individuals had an additional veterinary specialty degree (“Fachtierarzt/-ärztin”; two for laboratory animal science, one for equine, one for reptile, and one is currently finishing theirs for bovine). To maintain confidentiality, pseudonyms were used to identify participants in the tabulated list (see Table 1).

### 3.3. Collecting Data

The data collection process involved a comprehensive comparative approach that required a significant time commitment from the researcher to understand the various fields deeply. This involved immersion in the empirical field and an extensive literature review encompassing veterinary medicine, ethics, cultural studies, sociology, psychology, and ethology. This in-depth exploration was essential to adequately grasping the complexity and heterogeneity inherent in the research topic.

Semi-structured interviews served as the primary data collection method. The interviews, conducted between 2020 and 2022, ranged from 45 min to 2 h. Before the interviews, an informed consent form and a brief socio-demographic questionnaire were sent to the respondents, who were verbally informed about the study.

The interview guide focused on three key areas: daily killing work, professional and killing biography, as well as professional and social framework (see Appendix A). It was designed to reconstruct the “*interpretive knowledge*” of veterinarians [53]. By adopting an open-ended approach in the interview guide, the study encouraged participants to share diverse experiences of killing work in veterinary medicine. It fostered empirical openness and included biographical narratives, such as educational journeys and transitions between or simultaneous work in different areas of veterinary medicine. It also allowed the interviewees themselves to assess the relevance and boundaries. During the research process, the interview guide was iteratively modified according to the principles of grounded theory [46,52].

As part of the overall project, the interviews will be complemented by an in-depth analysis of relevant documents such as websites, guidelines, and manuals. However, they are not part of this article.

### 3.4. Analyzing Data

The coding procedure followed the constructivist grounded theory methodology outlined by Charmaz [46]. The interviews were transcribed, anonymized, and coded using MAXQDA 2022 software, providing initial empirical insights that informed subsequent data collection and deepened the understanding of the phenomenon of animal killing.

Data analysis began immediately after the first interview and involved a line-by-line approach, asking analytical questions of the data. The goal was to identify data segments (coding) that reflected the content and represented analytical abstractions. This initial coding and integration of existing literature (e.g., theoretical concepts) through deductive thinking led to identifying significant differences in motives, methods, locations, and fields. These findings guided deliberate decisions about further interview recruitment (theoretical sampling) and focused the direction of the study.

The analysis involved comparisons within and between the data and the codes using the constant comparative method [51]. Categorization, a crucial analytical step, focused on selecting codes that captured common or particularly significant themes and patterns derived from multiple codes. This process led to the development of a refined code system through two coding variants: initial coding and focused coding. Based on the researcher’s focus and prior analytical work, the latter facilitated a focused data exploration.

In addition to coding, the situation analysis (SitA) mapping strategies, according to Clarke et al. [54], were used to visualize the heterogeneous research field cartographically. A GT coding group and a SitA interpretation group promoted intersubjectivity in interpretation. The use of interpretation groups in this research project was instrumental in enriching the interpretation process and improving the quality of the analysis. By including multiple perspectives, the interdisciplinary interpretation groups facilitated the development of shared understandings and diverse interpretations of the data [55]. Within these groups, interview excerpts were openly discussed, allowing for collaborative exploration of analytical abstractions and the creation of analyses and models [55,56]. Notably, the author had sole responsibility for conceptualizing and writing the article and overall authority in the research process. This included making decisions about data collection, selecting passages for interpretation groups, and pursuing and rejecting analytical and conceptual ideas that were subsequently elaborated and incorporated into the final written work.

Writing memos complemented the coding process by capturing thoughts, data segments, codes, and relationships between codes in written form. The category system and memos were crucial in constructing an analytical conceptualization that was continually refined through data engagement. Integrating theoretical concepts as “*sensitising concepts*” (Blumer) guided exploring specific aspects, such as the ethical boundary work used in this article.

The analysis process involved a combination of inductive, deductive, and abductive methods to make connections between categories and construct explanatory diagrams that provided insights into the observed phenomena in the data. Throughout the research process, the researcher extensively documented emerging codes and categories and reflections on the research process through detailed memo writing.

### 3.5. Reflecting on Subjectivity

In addition, the dual positionality of the researcher as both a veterinarian and a sociologist presents both advantages and challenges to this study. On the one hand, this unique perspective provides valuable access to the field and a personal understanding of the subject matter. The researcher’s first-hand experience as a veterinarian can contribute to a deeper understanding of the nuances and complexities of veterinary practice. However, this dual role also raises concerns about the potential influence of personal experiences and biases on the research process and interpretations. To address this, extensive self-reflection and predescriptive interpretation groups were integral to the research process to mitigate potential bias and maintain a rigorous approach.

Despite these inherent challenges, the study design and methods employed in this research project provide a valuable contribution to the field of veterinary ethics. By acknowledging and actively addressing the potential impact of subjectivity, this research project strives to provide a comprehensive and well-rounded understanding of ethical considerations in veterinary medicine.

### 3.6. Limitations

It is essential to acknowledge the limitations of this study. First, it should be noted that this research is exploratory in nature, primarily due to the paucity of sociological research in veterinary medicine and the comparative research design employed [1,24]. And second, because of the small sample size for each field.

However, despite these limitations, this study utilizes rigorous methods such as constant comparison and theoretical sampling to contribute to theory development. It is important to note that representativeness is not a quality criterion within the GT framework [46,50,52]. Instead, the focus is on generating theoretical insights and developing a rich understanding of the phenomenon under study rather than achieving statistical representativeness through sample size or demographic characteristics.

While the comparative approach allows for an in-depth exploration of ethical boundary work in veterinary medicine and provides valuable insights into the complexities of the field. The findings of this study contribute to the existing body of knowledge and open avenues for further research and theoretical development in the field of veterinary ethics.

## 4. Empirical Findings: Reasoning and Ethical Boundary Work in Veterinary Medicine

In its empirical findings, this article explores how veterinarians use different forms of reasoning to define and justify euthanasia as “good”. Examining the differing justification narratives between the two fields will help identify the ethical boundaries veterinarians use to determine what constitutes a legitimate act of euthanasia.

Veterinarians face the complex task of balancing conflicting interests and responsibilities in their decision-making and legitimation processes to achieve a morally acceptable form of euthanasia. This article argues that veterinarians engage in ethical boundary work on two levels: within and between the two fields. First, within each domain, veterinarians establish internal ethical boundaries by using medical reasoning as the primary justification for euthanasia. This boundary allows them to present euthanasia as morally justifiable and distinct from other forms of killing.

However, euthanasia decisions in veterinary medicine are influenced not only by medical considerations but also by contextual factors. Because of the triadic structure of medical staff, clients, and animals, veterinarians incorporate the best interests of the client into the decision-making process and therefore evaluate emotional and economic reasons in addition to medical considerations when evaluating whether to euthanize an animal. For example, emotional reasoning occurs when clients are reluctant to part with an animal despite veterinary recommendations, while economic reasoning occurs when financial constraints affect the feasibility of continued care. These different forms of reasoning require a delicate balancing act that recognizes all parties’ interests.

This is where the second level of ethical boundary work comes into play as veterinarians navigate between different areas of veterinary medicine. They distinguish between caring for farm and companion animals and engage in comparative demarcation. Veterinarians actively contribute to the (re)production of societal distinctions related to emotional attachments and economic considerations associated with specific categories of animals. The interviews reveal instances in which veterinarians criticize and demarcate other fields, drawing attention to concerns such as over-commercialization in agriculture or over-treatment in small animal husbandry. These critiques and responses shed light on the complex ethical challenges veterinarians face and contribute to a spatial form of ethical boundary work within the veterinary profession.

Overall, the spatial ethical boundary work of veterinarians serves as a crucial strategy for legitimizing diverse moral positions and maintaining normative boundaries within the profession. However, empirical insights show that boundaries within the veterinary profession are inherently fluid rather than rigid and absolute. The complex dynamics of veterinary practice demonstrate that ethical boundaries are subject to constant negotiation and adaptation. Through this empirical approach, we can understand the nuanced and ever-evolving character of the profession’s moral infrastructure. By acknowledging the negotiation and fluidity of boundaries, we gain a deeper understanding of how veterinarians navigate diverse moral positions and engage in ethical deliberation to maintain their normative frameworks.

### 4.1. Within the Fields: Justifying Euthanasia Using Medical Reasoning

During interviews, the veterinarians consistently provided explanations and justifications for killing animals. The justifications were not always based on a full medical report. However, the justifications usually referred to medical reasons, such as an animal being critically ill. An example of this is a case reported by Vet. Treiber. During one of her internships, a dog was euthanized. She starts her narrative as follows:


*“An older family dog, just like a family member, and it was, the whole family was, uh, there and it was a home visit, so we went to the people’s house, and there we euthanised it and um I don’t remember anymore what it had, but it was anyway uh age-related.”*
[Vet Treiber]

For her, it was important to justify the killing, but not so much with a concrete, exact medical reason. Therefore, she refers to the explanation of the responsible veterinarian:


*“Anyway, she explained the case to me. And with it a little bit, uh, justified why we do it: so that the dog is somehow chronically ill, and has no more life worth living, and that it has a long medical history and that somehow there is no other therapy.”*
[Vet Treiber]

As shown above, the explanation is based on the dog’s “*long medical history*” and chronic illness without further specifying the dog’s medical condition. The necessity is presented to her via the lack of medical alternatives (“*somehow there is no other therapy*”) and the assumed interest of the animal (“*a life worth living*”).

The justification narratives involve assessing whether an animal is in pain and suffering and has no quality of life. These classic physiological-pathological reasons point to a necessity where death is presented as the only alternative or better solution. As Vet. Treiber puts it, “*This is a release from a life that I no longer find worth living*”.

But how do veterinarians understand the concept of “*release*”? And how does this discourse become medical? The narratives of euthanasia have two dimensions: (1) the purported interest of the animal and (2) the medical necessity of the action. The justification narratives involve assessing whether an animal is in pain and suffering and has no quality of life. As such, it is embedded in a medical discourse that relates to medical actions such as diagnosis, prognosis, and deciding on a therapeutic goal and treatment. For example, Dr. Bauer, a veterinarian working in companion and farm animal medicine, states, “*Now would be the time to euthanize the animal before there is more suffering*”. Therefore, she justifies the killing with (1) the best interest of the supervised animal (“*more suffering*”) and (2) her particular task as a veterinarian of determining, classifying (“*greater*”), and evaluating (“*what I … feel*”) the animal’s interest and suffering. These classic physiological-pathological considerations are linked to a moral necessity in which death is presented as the only alternative or better solution. Thus, veterinarian Dr. Kaufmann’s concept of “*absolute release*” emphasizes that euthanasia is a necessary evil that liberates the animal with a “good” death.

The release narrative is found in both companion and farm animal medicine. For example, Dr. Bauer describes a situation with a horse in which euthanasia is necessary due to a proximal leg fracture that cannot be treated: “*There are irreparable injuries, such as a proximal leg fracture in a horse, where no therapy is possible. Um, where the horse must be put out of its misery*”. Similarly, she mentions euthanasia in the case of critically ill dairy cows: “*with the dairy cows, or something like that when they were critically ill, and there was no hope of recovery*”. In both companion and farm medicine, the release narrative refers to medical necessity in addition to alleged animal interests, and euthanasia is seen as necessary in the case of “*hopeless injuries*” or a “*critically ill*” animal with “*no hope of recovery*”.

In summary, medical reasoning is a justification strategy embedded in a “*release*” narrative. Veterinarians legitimize euthanasia when it is medically necessary and in the animal’s best interest. But the decision-making process involves stakeholders other than the animal and the veterinarian, such as the client and the government. In the next part, the article explores what happens when a conflict of interest occurs, particularly with the client, and how veterinarians use this as an explanatory reason (emotional and economic reasoning).

### 4.2. Between the Fields: Explaining Euthanasia Using Emotional and Economic Reasoning

Veterinary medicine has some peculiarities compared to other health professions. One is the way the work is organized. In Germany, for example, many veterinarians are still self-employed, and there is no national health insurance system as there is in human medicine. As a result, veterinarians refer to themselves as entrepreneurs [57,58]. Currently, private animal health insurance is still the exception in Germany, and because of the triadic structure, the client’s needs are important for the business. Veterinarians’ narratives, therefore, include the client’s perspective. In negotiating an animal’s euthanasia, veterinarians include economic and emotional considerations and, in turn, draw the boundaries of “good” killing. Here, the differences between the two fields are striking.

For example, Dr. Schmidt, a farm animal veterinarian, describes how he sometimes must euthanize a farmer’s dog. For him, there is a significant difference between this and what he considers his daily euthanasia work (euthanizing a cow):


*“But of course, these are dogs that I have known for a long time. And they are usually euthanised because of their old age. (…) And these are then also euthanasia, where the farmer has already prepared for it, so to speak. Because he knows: ‘Oh, the dog, he won’t make it! And one way or another, it must stop.’ That’s basically more difficult with dogs, uh, to somehow get the owner to realise that the animal is so, that this is no longer a life for the dog.”*
[Dr. Schmidt]

Dr. Schmidt focuses on the animal and its quality of life: “*that it is no longer a life for the dog*”. This is part of the medical reasoning described above, but he bases the difference on another aspect: the (more) emotional bond between the farmer and the animal, which makes the decision “*harder*” for the farmer.

This narrative reflects the role of veterinarians in (re)producing the ambivalent classification of animals as close companions and even “*family members*” [Vet Treiber] in companion animal medicine while being perceived as food producing livestock (in German law as Lebensmittellieferndes Tier) in the context of farm animal medicine [17,18,25]. This juxtaposition of emotions and economics significantly diverges between the two fields. However, we must delve into the veterinarians’ narratives to fully understand these dynamics.

#### 4.2.1. Economic Reasoning: Distinguishing Farm Animal Medicine

Economic reasoning plays a critical role in managing conflicts between the interests of clients and animals in veterinary practice. While medical reasoning is central to both fields, economic reasoning is valued differently. This divergence raises thought-provoking questions and highlights the spatial dynamics of ethical boundary work between fields.

Veterinary medicine is influenced by socio-historical changes in human-animal relationships [25]. Dr. Kaufmann highlights a recent shift in client attitudes toward horse euthanasia:


*“It is, uh, it is actually, so nowadays that one is usually uh earlier in the, inclined to break off than the animal owners, so according to the motto: ‘Can’t you do anything else? And we want to have done everything,’ whereas in the past, uh, from time to time, uh, so maybe 20, 25 years ago, uh, sometimes, uh, they would try urgently, uh, to euthanise the animal. One would see that it has no sense anymore. Preferably with riding horses, which, if they can no longer be used for riding, are uh simply also uh yes expensive, uh no longer have any sporting value. The owner can no longer do anything with it, but the horse may stand on the pasture for another five or ten years and eat, but it will always cause veterinary costs. Uh, and then one is asked from time to time, uh, nevertheless, that one may nevertheless realise that it can’t continue anymore. But these are, of course, pretended economic reasons for the animal owner. And we actually reject such things. And then, uh, it also happened that we, I say we in the plural, because my colleague uh there uh actually holds the same opinion as I do, uh, so we also reject the one or the other order.”*
[Dr. Kaufmann]

In the past, clients occasionally requested the euthanasia of horses that no longer served a sporting purpose or had diminished economic value. However, as horses have transitioned from being perceived as mere farm animals to highly valued companion animals, the evaluation of economic factors has taken on a different meaning [18,25]. Dr. Kaufmann and his colleague *(*“*we also in the plural*”) strongly discourage requests for euthanasia based solely on the client’s economic considerations. This example draws attention to two critical points: the influence of economic factors on animal utility and the different ethical boundaries drawn depending on how animals are categorized.

Vet. Treiber, another companion animal veterinarian, connects agriculture and economics. For her, this is exemplified by the differences in the decision-making process between farm and companion animals. She has no moral problem with killing an animal for the right reasons (see Section 4.1). Still, she sees it differently when it is an economic decision: “*but what I judge differently is killing for economic reasons*”. She goes on to describe her experience during an internship in the field:


*“It happened that we had cattle that, uh, were not terminally ill, as I know it from the small animal sector. But once there was a cow that had a severe, how do I know, an injury to the carpal joint and could no longer stand. And there is, of course, no surgical intervention planned that costs, how do I know, several hundred euros; and takes a long, uh, time to heal. But there, the costs are weighted: ‘This cow doesn’t give so much milk, anyway; that goes away.’ And that was for me after a while, of course, also normal: It’s a farmer who lives from it. And that is just an economic area. But as a vet, I find that really difficult, and I realise that that wouldn’t be my area. So I won’t make that decision and support it, … I found this economy in that area of, uh, that is difficult for me.”*
[Vet. Treiber]

This quote shows Vet Treiber criticizing the field because of the importance of economic constraints and drawing a line at “good” killing when economic circumstances prevail. She describes this experience as a reason not to work in the field. For her, it is inconsistent with her understanding of what it means to be a veterinarian (“*as a vet*”).

Dr. Fischer, a farm animal veterinarian, describes the same phenomenon but with a different normative evaluation. She focuses on economic considerations and their impact on her field’s diagnostic scope and decision-making process. By examining Dr. Fischer’s perspective, we can gain insight into how ethical boundaries are drawn differently in these practices:


*“And, of course, for economic reasons, my diagnostic scope in farm animal practice is quite different from that of my small animal or equine colleagues. And yes. And, of course, and this has to be clearly said, economic efficiency also plays a role! As I said, it is not the case that you build an external fixator for a cow with a femur fracture, for example, and then try to mobilise it for eight weeks in an aqua trainer. The animal is just not worth it. And I always find it a little bit unfair, and I have to say because it’s often not the reality that people are so quick to blame the farmers that they say, ‘Yeah, uh, they put them to sleep anyway, no animal is being treated there anyway.’ That’s not true! But the decision to euthanise is made more quickly but no less consciously. After all, if it’s possible to treat, it will be worthwhile for the farmer. In principle, euthanasia is never worthwhile for the farmer in the end.”*
[Dr. Fischer]

Dr. Fischer emphasizes how financial constraints influence the diagnostic scope, as the cost of treatment must be weighed against the animal’s economic value. She points out that farmers will consider treatment worthwhile if it is promising and economically viable, while euthanasia is often considered financially unfeasible given the previous investment without a corresponding return. Due to the economic origin of agriculture, this economic factor is generally discussed much more openly in interviews with farm animal veterinarians. This was not the case with the companion animal veterinarians interviewed.

By examining the intersection of economic reasoning and ethical boundaries in veterinary medicine, particularly in relation to euthanasia, we gain insight into how veterinarians draw these lines. In farm animal medicine, the value of the animal is obviously tied to the client’s economic interests. However, the extent to which the client’s economic interest justifies killing is debated differently in the two fields of veterinary medicine.

#### 4.2.2. Emotional Reasoning: Distinguishing Companion Animal Medicine

Emotional reasoning plays a vital role in veterinarians’ ethical boundary work. As shown, farm animal veterinarian Dr. Fischer negotiates an ethical boundary between economic and emotional justifications. To be a “good” killing (in the normative sense), actions are explained not only by financial but also by the emotional interests of clients. Emotions and how to deal with them are of central importance to veterinarians, especially when there is a close emotional bond between the client and the animal, which is often the case in small animal medicine [19]. But how is the legitimacy of emotional reasoning negotiated?

Some veterinarians are concerned about the excessive emotionalization and subsequent overuse of medical interventions in companion animal medicine. For them, the situation raises ethical questions about whether animals are being kept alive primarily to satisfy the client’s emotional needs rather than to prioritize the animals’ interests. These critiques highlight a departure from the concept of “good” killing. By exploring how emotional reasoning is critiqued and legitimized in companion animal medicine, we can better understand the spatial ethical boundaries around euthanasia. For example, Dr. Wagner, a former farm animal veterinarian who now works in a slaughterhouse, critiques the following:


*“Veterinary medicine has some of the same standards as human medicine. A vet could probably give Grandma a new hip. But first of all, I think it’s a bit bad that in a world where people are starving, you do something like that, that you spend thousands of euros on an animal that doesn’t get much better afterwards, because, I mean, then it has a new hip, then its shoulder hurts at some point. An old animal is just old, and you must do to it what you do to us old people. It would be best to keep an animal alive until it dies of old age, and it must fight for years with some things; I don’t know what. And then they get painkillers, of course, but I know how it is with broken bones. (…) And I think it’s a sin to torture animals so that people don’t have to say goodbye to them. Yeah, what are you doing to the animals? That’s all sentimentality, and that comes from the fact that people have a nonsensical fear of death, so if you have a relaxed relationship with death, you can send your animal to its death. You can say, ‘Guy, you can’t go on; you can go now.’”*
[Dr. Wagner]

She critically views the ethical justification for euthanasia in companion animal medicine on the grounds of “*sentimentality*” and the prioritization of the client over animal interests. She questions allocating resources for extensive medical interventions on animals when humans are starving. For her, prolonging an animal’s life in the face of suffering is unnecessary and even cruel because of societal fear of death. Instead, a peaceful and timely euthanasia decision is more important to them for the “good” end. And a social opportunity for an accepting attitude towards death would be beneficial. This critique challenges the ethical boundaries of companion animal practice, particularly regarding emotional reasoning in euthanasia decisions.

Dr. Fisher’s critique is consistent with the earlier discussion and emphasizes the need to balance the interests of the animal with those of the client:


*“Whereby I also have to say, when I sometimes see what our small animal colleagues do, uh, I am not always convinced whether this is still in the best interest of the animal. When, as I recently experienced when I was in practice, a dog with a liver tumour and metastases is kept alive for ten days because someone must come back from vacation or something, (1) then I have a stomachache, too. I have to say, ‘I’m not satisfied.’”*
[Dr. Fischer]

Here, Dr. Fischer questions whether certain treatments are really in the animal’s best interest. For example, she recounts the case of a terminally ill dog that was kept alive simply because someone had to return from vacation. While Dr. Fischer finds these cases troubling, she does not classify them as ethical issues but rather as personal opinions or observations. In sum, the critique of farm animal veterinarians highlights instances where “*sentimentality*” (Dr. Wagner) and client preferences can override the best interest principle.

But what about emotional reasoning and its associated ethical boundaries from a companion animal perspective? Dr. Sander, who has experience working in a large research facility and assisting in a small animal practice, highlights the evolving perception of companion animals (cats, dogs, and horses) and the advances in medical technology in these areas. This sheds light on the ethical boundary work involved in justifying euthanasia and the changing dynamics surrounding the emotional connection to companion animals:


*“So, the way away from the animal as a thing, even if, let’s say, in the mid-1980s, the animal may have had a high value for the individual owner of, let’s say, the clinic, I don’t think you can compare that with what’s happening today. It starts with the whole range of equipment available for examinations, where people pay six or 700 euros for an MRI of their dog or horse, even if it’s hard for them, yes? I think animals are seen more as full family members than they used to be. In the past, but I can only speak from my own experience, people dealt with it more professionally. According to the motto, the dog will be nine, ten or twelve years old; if you’re lucky, it’s over, and then it’s euthanised, right? And the therapy options were not like that either. When I started, people were more willing to put a dog or cat to sleep before they would try again.”*
[Dr. Sander]

The increasing recognition of animals as valued “*family members*” has influenced the way euthanasia is approached [19,25,31]. There was a more pragmatic view in the past where people accepted that animals had a relatively short lifespan and euthanasia was an expected outcome. In addition, the therapeutic options available were limited compared to today. However, the emotional bond between clients and their animals has led to a greater willingness to explore various treatment options before considering euthanasia.

Dr. Sander’s observations highlight the spatial dimension of ethical boundary work in veterinary medicine. The changing perception of companion animals, from objects to cherished family members, has led to new considerations and expectations regarding their care, treatment, and death in this field. The increased availability of advanced medical technologies further blurs ethical boundaries as clients now have more treatment options to explore. Veterinarians must balance honoring the emotional bond between animals and clients with ensuring the animal’s interests. Ethical boundary work is critical in determining the most compassionate and responsible course of action for companion animals, considering their evolving status and advances in medical technology.

Ethical boundary work becomes particularly relevant when a client refuses to euthanize an animal despite the veterinarian’s belief that euthanasia is necessary. In such cases, veterinarians may need to provide emotional support and help clients understand the situation. An example is Dr. Grimm’s account of a horse with central nervous system problems that caused it to exhibit self-destructive behavior by circling and repeatedly crashing into walls. Despite the veterinarian’s recommendation for euthanasia, the horse’s owners initially refused. To address this situation, Dr. Grimm arranged for the horse to be moved to an open stall where it could move freely. When the clients observed that the horse was not coping well in the new environment, they finally agreed to euthanasia.


*“And then I said, ‘Well, now she has the best possible accommodation,’ then I went to this open stall and said, ‘And how is she?’ ‘Yes, it’s impossible; she doesn’t know where the water is!’ No, like that. And then they saw that first-hand and were convinced that you couldn’t save this horse. Yes, but yes, that, that was a lucky coincidence, but that’s sometimes more difficult, yes. But usually, it’s like that. But if they don’t want to hear that, then they don’t call you anymore, and they call somebody else.”*
[Dr. Grimm]

This example illustrates the importance of providing alternative solutions and facilitating a better understanding of the animal’s condition and well-being. It shows that veterinarians can help clients make more informed decisions in the animal’s best interest through emotional support, practical interventions, and open communication. However, Dr. Grimm also acknowledges the challenges in such cases. Sometimes clients are not open to considering euthanasia or accepting the veterinarian’s recommendations. In these situations, they may seek another veterinarian’s opinion or terminate the relationship altogether. This highlights the complexity of navigating the intersection of emotions, ethical boundaries, and the client-veterinarian relationship.

These contrasting perspectives highlight the complexity of ethical boundary work regarding emotional reasoning. While advocating a compassionate approach that upholds the animal’s best interests, some critics in veterinary medicine caution against the potential overuse or misuse of interventions driven solely by “*sentimentality*” (Dr. Wagner) or client demands. Veterinarians have to deal with the emotional relationship between clients and animals. At the same time, they must put the animal’s welfare first. The goal is to balance emotional and medical considerations.

In conclusion, veterinarians process emotional and economic justifications in addition to medical ones when drawing boundaries around what constitutes “good” killing. It was possible to show how spatial ethical boundary work was used in the interviews. Companion animal veterinarians criticized the overemphasis on the economic interests of clients. On the other hand, the farm animal veterinarians criticized the boundary extension as being too emotional. This requires veterinarians to skillfully navigate ethical boundaries and balance emotional compassion, financial constraints, and medical reasons to keep the practice within the bounds of “good” killing.

Understanding these different viewpoints and the spatial ethical boundary work within veterinary medicine is essential to understanding the moral infrastructure. It enables the profession and society at large to reflect ethically on the decision-making processes and legitimacy of animal euthanasia.

## 5. Critical Reflection: The Empirical Messiness of Boundaries

This article addresses the topic of ethical boundary work in veterinary practice. It focuses on euthanasia in companion and farm animal medicine and provides a valuable understanding of the complex dynamics involved. While there are variations in context, location, and stakeholders (especially differences in clients), the method (injection) and purpose (to end suffering) are the same in all cases discussed [3,13]. Further research is needed to include additional forms of killing.

In summary, veterinarians consider both the animal and the client as objects of care, which leads them to consider not only medical justifications but also economic and emotional reasons in their decision-making. The extent to which these factors should be considered becomes part of the negotiation within and between different fields of veterinary medicine. This juxtaposition leads to significant differences between the two fields. Therefore, veterinarians perform ethical boundary work on two levels. First, veterinarians establish internal ethical boundaries within their profession by using medical reasoning as the primary justification for euthanasia. This allows them to distinguish euthanasia from other forms of killing and present it as morally justifiable. Second, veterinarians engage in boundary work between different fields by using economic and emotional reasoning to explain and justify the ethical boundaries in their own field while criticizing the other (spatial ethical boundary work).

These two lines of demarcation can be seen hand in hand. They reflect the ambivalent relationship between humans and animals today. The different norms are seen as part of the profession through the spatial form of critique and counter-critique, and a constant negotiation about “good” killing occurs. This happens not only in the narrative of each veterinarian but also in the profession as a whole. The spatial ethical boundary work of veterinarians is crucial for legitimizing different moral positions and maintaining normative boundaries within the profession.

It is important to note that this article’s narrative reflects societal expectations of farm versus companion animal categorization by omitting the relevance of economic considerations in small animal medicine (e.g., in the case of inability to pay or even “convenience euthanasia”). The situation is even more complex, as emotions and finances play a role in both fields [19,59,60]. However, interviewees do not equally address them. For example, farm animal veterinarians generally discuss economic factors more openly than their companion animal counterparts. The categorization of animals and related societal expectations are confronted with empirical ambiguity. This becomes more complex when other uses of animals and areas of veterinary medicine are added, e.g., wildlife in conservation medicine and laboratory animals in laboratory animal medicine. Furthermore, it is essential to recognize that veterinarians’ understanding of euthanasia cases may differ from that of philosophers and ethicists [23,59]. Distinguishing between justifying (medical) and explaining (emotional and economic) factors in an ethical sense is crucial [7], but further research is needed to study how veterinarians perceive and use these factors in practice, which could be conducted through methods such as ethnographic research.

Finally, this discussion highlights the importance of openness in empirical study design. We gain valuable insights into this complex process by examining ethical boundary work in practice. In practice, clear boundaries are often absent. As empirical observations show, veterinary medicine boundaries are inherently messy [17,18]. Disciplinary boundaries are fluid, with individuals working in different areas simultaneously or throughout their careers, animals appearing in various roles (research, agriculture, and companion animals), and location adding further complexity. The complexity of ethics in practice transcends the confines of discipline-specific research or study designs, necessitating empirical research that fully acknowledges and embraces this complexity.

In summary, this article argues for a more open approach that explores the moral and ethical challenges within the veterinary profession while incorporating the perspectives and knowledge of veterinarians. Such an approach allows for a deeper understanding of the moral infrastructure of the profession and its implications for practice.

## 6. Conclusions

In conclusion, this article empirically explores ethical boundary work within veterinary medicine, focusing on euthanasia practices in companion and farm animal medicine. For veterinarians, euthanasia is both a routine practice and an issue that raises significant ethical questions in light of the changing human-animal relationship. By investigating the criteria and levels that shape these intra-professional boundaries, the study highlights the diverse moral landscape within veterinary medicine. The interplay between forms of reasoning and ethical boundary work is crucial to understanding veterinary practice.

Veterinary ethics is challenged by the diversity and empirical messiness of veterinary practice, where animals are used in different ways and take several social-moral positions in modern society. In addition, veterinarians have multiple objects of care to consider, including the animal, the client, the broader context, and themselves. The empirical exploration of ethical boundary work in euthanasia practice allows the moral infrastructure of veterinary medicine to be subjected to social discussion and normative evaluation. However, the findings presented here represent only the beginning of studying this complex topic. Further research and ongoing dialogue are needed to advance our understanding of ethical boundary work in veterinary medicine and its implications in practice.

## Figures and Tables

**Table 1 animals-13-02515-t001:** Information about the sample.

Nr.	Pseudonym	Gender (f = Female, m = Male)	Field of Work (Ranking of Fields According to Relevance)	Years of Experience
01	Kathrin Diehl	f	Student Small Animals	Junior
02	Antonia Reuter	f	Research (Regenerative Medicine) Equines Small Animals	Junior
03	Mara Treiber	f	Research (Anatomy) Small Animals	Junior
04	Linda Müller	f	Laboratory Animals	Junior
05	Anika Bauer	f	Small Animals Equines Farm Animals	Intermed.
06	Beate Grimm	f	Equines Small Animals	Intermed.
07	Hanna Fischer	f	Farm Animals (Bovine)	Intermed.
08	Lars Schmidt	m	Farm Animals (Bovine)	Intermed.
09	Pippa Tamme	f	Farm Animals (Bovine)	Intermed.
10	Sandra Ording	f	Farm Animals (Bovine)	Intermed.
11	Nadine Mahr	f	Veterinary Office Small Animals	Intermed.
12	Claudia Nagel	f	Laboratory Animals	Intermed.
13	Michael Kaufmann	m	Small Animals Equines	Senior
14	Niklas Zinn	m	Equines	Senior
15	Anette Mayer	f	Laboratory Animals Small Animals	Senior
16	Marion Arnold	f	Exotic Animals (Reptiles)	Senior
17	Emma Wagner	f	Veterinary Office Farm Animal	Senior

## Data Availability

The anonymized data supporting the conclusions of this article are provided by the author.

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
