# Peer review of "Justifying Euthanasia: A Qualitative Study of Veterinarians’ Ethical Boundary Work of “Good” Killing"

_animals, 2023, doi:10.3390/ani13152515_

Round 1

Reviewer 1 Report

Thank you very much for the article on this highly relevant topic. It was a pleasure to read it.

Summary: The article presents a qualitative interview study with veterinarians from different fields (small animal and farm animal). The focus was on the ethical tensions and challenges regarding euthanasia, in particular the comparison between the two fields regarding that matter. The author used a Grounded Theory approach for conducting and analysing the interviews, and, furthermore, used the framework of ethical boundary-work to elaborate on the veterinarians’ conception of “good killing”. The study revealed that there are differences in those conceptions among veterinarians, particularly when dealing with/talking about farm vs. companion animals. Additionally, emotions and economic factors were found to be influential in the decision-making processes concerning euthanasia. The author concludes that applying the framework of ethical boundary-work helped to identify the complex ethically challenging aspects of killing animals in veterinary practice.

General comments:

The author is investigating a much debated and – given the changing role of animals in our society – increasingly important topic in veterinary medicine.

Concerning the methods:

As I don’t have a background in social sciences I felt well informed about the framework of ethical boundary-work in the second chapter and I can see why it makes sense to apply it in this study. Having said that, I would like to read some more details about the methods in the third chapter. Not all readers of “Animals” might be familiar with the constructivist grounded theory coding methodology by Charmaz, so please summarise the procedure. I also wondered if the mentioned visualisation (see ll. 192-195) is part of the manuscript. Furthermore, you mention a coding group and an interpretation group. However, this is a single author paper, so I would like to learn more about the collaboration and the participation in the coding process.

I understand that it was a theoretical sampling but could you write one more sentence about how the participants were identified and recruited? Could you provide the interview guide as a supplement?

A huge strength, in my opinion, is the self-reflection about your own role and the influence of your interdisciplinary background (ll. 218-227) which is pivotal in qualitative research and in particular if the research is mainly conducted by one author.

Results:

The results were presented convincingly and the participants’ statements supported the analytical passages quite well. To me, the summarising paragraph at the beginning of the fourth chapter really highlights the most important findings of the analysis. In the sub-chapters, the paraphrasing of the statements is sometimes a bit repetitive (see, e.g.,ll. 394-416 where you first announce a paragraph on economic considerations and their impact on the diagnostic scope etc., then you give the statement on that and afterwards you paraphrase the statement, repeating most of what the participant said.; similarly, in ll. 486-497), but I am aware of the fact that this cannot always be avoided in reporting qualitative research.

The Discussion and Conclusion are, in my opinion, well-written. I would like to add one aspect that you might want to include, though. The difference between the animal’s best interest to put an end to their suffering and the client’s interest to prolong the animal’s life (for emotional or economic reasons) is well-debated. However, the additional aspect that death can mean a harm to the animal (or maybe it can’t?) also presents an aspect of the factor-weighing and decision-making process that veterinarians might consider. In a way, there are competing and not directly accessible interests (end suffering; continue to live) within the animal that come on top of the potentially conflicting interests of further stakeholders and that present a challenge to veterinarians. Maybe you found something on that aspect in your interview data?

Further comments:

l. 81: should it be “human euthanasia” instead of “human medicine”?

l. 119: The sentence “a great deal of action occurs at the borders” is not clear to me. Could you rephrase that?

l. 170: It is unclear to me if “this study” refers to Anderson and Hobson-West or to your study

l. 185: Please clarify if the participants were from Germany (which I assume). It might be important as the legislation on euthanasia and veterinary education differ between countries

l. 190: should it be “45 minutes and 2 hours”?

ll. 198-199: If I understand it correctly, the further documents are part of the sociological project but not of this particular interview study/manuscript?

l. 275: I assume, the veterinarian Treiber was the intern in this statement and not yet a veterinarian? Maybe you could describe that more clearly?

ll. 342, 343: Are the terms in quotation marks taken from the participants’ statements or quoted from the literature?

l. 366: please rephrase “that the now longer goes”. I assume that this is a translation problem but maybe the statement can be phrased more meaningfully?

l. 391: This is a very interesting statement. If I understand correctly, she says that she decided against the field of farm animal veterinary medicine (at least partially) due to the fact that decision-making is based on economic factors, there. Maybe you want to add a comment on that?

Reviewer 3 Report

The Authors investigate the euthanasia practices in companion and farm animal medicine and the ethical boundary-work that veterinarians do.

The aims presented in this article could be very interesting for the scope of the journal, but the manuscript lacks objectivity, rigor and reproducibility.

L183-184: “This study used a grounded theory (GT) approach to examine the ethical boundary-work between fields of veterinary medicine comprehensively”  I think that it is necessary for the better understanding by journal's readers, to give a short explanation of the grounded theory and to cite references.

Expert interviews were conducted with a low number of veterinarians (only 17) from different disciplines and specializations and it leads differences in motives and methods quickly emerged and difficult to compare. These interviews were reported without a scientific method and/or an elaboration or statistically analysis.

I suggest increasing the number of respondents, statistically analyzing the interviews and to elaborate graphics and/or tables in order to better explain how ethical boundary-work is practiced in veterinary medicine and to provide insight into the challenges and tensions that arise at these boundaries.

Although it is a potentially a very useful paper, especially from an ethical perspective, as currently written, it is of poor quality and is not a high-quality scientific report. So, for the reasons stated above, it cannot be recommended for publication in its present form.
